# High G2M Pathway Score Pancreatic Cancer is Associated with Worse Survival, Particularly after Margin-Positive (R1 or R2) Resection

**DOI:** 10.3390/cancers12102871

**Published:** 2020-10-06

**Authors:** Masanori Oshi, Stephanie Newman, Yoshihisa Tokumaru, Li Yan, Ryusei Matsuyama, Itaru Endo, Matthew H. G. Katz, Kazuaki Takabe

**Affiliations:** 1Department of Gastroenterological Surgery, Yokohama City University Graduate School of Medicine, Yokohama 236-0004, Japan; Masanori.Oshi@RoswellPark.org (M.O.); ryusei@yokohama-cu.ac.jp (R.M.); endoit@med.yokohama-cu.ac.jp (I.E.); 2Breast Surgery, Department of Surgical Oncology, Roswell Park Comprehensive Cancer Center, Buffalo, NY 14263, USA; snewman5@buffalo.edu (S.N.); Yoshihisa.Tokumaru@roswellpark.org (Y.T.); 3Department of Surgery, Jacobs School of Medicine and Biomedical Sciences, State University of New York, Buffalo, NY 14263, USA; 4Department of Surgical Oncology, Graduate School of Medicine, Gifu University, 1-1 Yanagido, Gifu 501-1194, Japan; 5Department of Biostatistics & Bioinformatics, Roswell Park Comprehensive Cancer Center, Buffalo, NY 14263, USA; li.yan@roswellpark.org; 6Department of Surgical Oncology, The University of Texas MD Anderson Cancer Center, Houston, TX 77030, USA; mhgkatz@mdanderson.org; 7Division of Digestive and General Surgery, Niigata University Graduate School of Medical and Dental Sciences, Niigata 951-8520, Japan; 8Department of Breast Surgery, Fukushima Medical University School of Medicine, Fukushima 960-1295, Japan; 9Department of Breast Surgery and Oncology, Tokyo Medical University, Tokyo 160-8402, Japan

**Keywords:** biomarker, cell cycle, gene set, pathway analysis, pancreatic cancer, tumor gene expression

## Abstract

**Simple Summary:**

The G2M checkpoint pathway is a critical step in the cell cycle, which is one of the hallmarks in cancer. We investigated clinical relevance of the G2M pathway in pancreatic cancer, using gene set variation analysis (GSVA) with Hallmark G2M checkpoint gene set. High G2M score tumors enriched several cell proliferation gene sets as well as *MKI67* expression, pathological grade, and proliferation score. Furthermore, the score was predictive of disease-specific survival in pancreatic cancer. High G2M tumor was associated with high mutation rate of *KRAS* and *TP53*, and significantly enriched these pathway gene sets, as well as high infiltration of Th2 cells. High G2M score consistently associated with worse overall survival in multiple cohorts, particularly in R1/2 resection, but not in R0.

**Abstract:**

Pancreatic cancer is highly mortal due to uncontrolled cell proliferation. The G2M checkpoint pathway is an essential part of the cell cycle. We hypothesized that a high G2M pathway score is associated with cell proliferation and worse survival in pancreatic cancer patients. Gene set variation analysis using the Hallmark G2M checkpoint gene set was used as a score to analyze a total of 390 human pancreatic cancer patients from 3 cohorts (TCGA, GSE62452, GSE57495). High G2M score tumors enriched other cell proliferation genes sets as well as *MKI67* expression, pathological grade, and proliferation score. Independent of other prognostic factors, G2M score was predictive of disease-specific survival in pancreatic cancer. High G2M tumor was associated with high mutation rate of *KRAS* and *TP53* and significantly enriched these pathway gene sets, as well as high infiltration of Th2 cells. High G2M score consistently associated with worse overall survival in 3 cohorts, particularly in R1/2 resection, but not in R0. High G2M tumor in R1/2 highly enriched metabolic and cellular components’ gene sets compared to R0. To our knowledge, this is the first study to use gene set variation analysis as a score to examine the clinical relevancy of the G2M pathway in pancreatic cancer.

## 1. Introduction

The rising incidence and high mortality rate make pancreatic cancer a clear public health threat [1]. The prognosis for pancreatic cancer is one of the worst among all types of cancer, with a 5-year survival rate of 6% [2]. Recent identification of borderline resectability allowed pancreatic surgeons appropriate patient selection to perform complete curative resection with microscopic tumor clearance (R0), which significantly improved median survival of these patients [3]. On the other hand, it is thought that pancreatic cancer that results in margin-positive after resection (R1 and R2) is more likely due to aggressive biology and invasiveness of cancer rather than surgical technique in the majority of cases [4,5,6]. To this end, a measure that can distinguish pancreatic cancer with aggressive biology is expected to be clinically useful.

It is known that several signaling pathways are involved in pancreatic cancer progression and clinical outcome [7,8,9]. Aberrant cell proliferation, possibly the most heavily studied hallmark of cancer, is mediated in part by alteration of cell cycle progression genes. One of the key components of cell cycle is the G2M checkpoint, which controls transition into mitosis through the action of cyclin B-cdc2 (*CDK1*) complexes [10]. The cells do not initiate mitosis until damaged or incompletely replicated DNA is sufficiently repaired. Several therapies focus on inactivating the checkpoint in order to force cells with excess DNA damage to proceed through mitosis and induce cell death. For example, the taxanes arrest cells at the G2M phase of the cell cycle that demonstrated significant impact against many solid tumors, such as gastric cancer [11], lung cancer [12], breast cancer [13,14], and pancreatic cancer [15,16]. Thus, we sought to quantify the amount of G2M checkpoint pathway activity.

The gene set variation analysis (GSVA) is a computational algorithm which allows us to examine gene expression within a pathway rather than a single gene. Utilization of a pathway approach more accurately reflects gene coordination, increases model simplicity, and can increase the applicability of prediction models. Indeed, we have previously developed the G2M pathway score using the Molecular Signatures Database (MSigDb) Hallmark G2M checkpoint gene set of the GSVA and demonstrated its utility as a prognostic biomarker of metastasis in estrogen receptor-positive breast cancer [17]. G2M score correlated with the infiltration of both pro- and anti-cancerous immune cells in primary as well as metastatic breast cancer, was associated with pathological complete response to neoadjuvant chemotherapy in estrogen receptor (ER)-positive/human epidermal receptor 2 (HER2)-negative breast cancer, and was predictive of response to cyclin-dependent kinase inhibition therapy. In the current study, we used the “HALLMARK_G2M checkpoint” gene set of the MSigDb Hallmark GSVA as the G2M checkpoint score in a pancreatic cancer cohort. Median value was used to divide the low and high G2M score group in each cohort. We hypothesized that the upregulation of the G2M pathway is associated with aggressive cancer biology and worse survival in patients with pancreatic cancer. We used G2M pathway score in pancreatic cancer tumors and examined whether it reflects cell proliferation both biologically and clinically, associates with known gene alterations, associates with immune cell infiltration, and associates with worse survival in pancreatic cancer.

## 2. Results

### 2.1. High G2M Pathway Score Reflects Increased Cell Proliferation in Pancreatic Cancer

We defined the G2M pathway scores as the Molecular Signatures Database (MSigDb) Hallmark G2M checkpoint gene set of the Gene set validation analysis (GSVA) that analyzes 200 genes, as we described previously (Appendix A, [17]). The median value was used to divide into High and Low G2M pathway score groups. Following our findings in breast cancer [17], we expected that G2M score would reflect cell proliferation of pancreatic cancer as well. Gene Set Enrichment Analysis (GSEA) demonstrated significant enrichment of *E2F* target, *MYC* targets v1 and v2, mitotic spindle, *MTORC1* signaling, *PI3K AKT MTOR*, DNA repair, and *p53* pathway gene sets to high G2M pathway score tumors in The Cancer Genome Atlas (TCGA) pancreatic cancer cohort (TCGA-Pancreatic-Adenocarcinoma [PAAD], *n* = 176) (Figure 1). These enrichments were mirrored in another independent pancreatic cancer cohort (GSE62452, *n* = 69). The percentage of overlapping genes were greatest between G2M checkpoint and *E2F* pathways (36.5%), and the overlap was less than 20% for the other gene sets (Appendix A). Combined, these results suggest that G2M pathway scores reflect cell proliferation in pancreatic cancer.

### 2.2. G2M Pathway Score Associates with Clinical Parameters of Cell Proliferation in Pancreatic Cancer

Cancer cell proliferation is commonly assessed by *Ki67* staining in clinical practice. Gene expression of *Ki67* (gene name: *MKI67*) was found to be significantly correlated with G2M pathway score in both TCGA and GSE62452 cohorts (Figure 2A; spearman *r* = 0.841 (*p* < 0.01) and *r* = 0.825 (*p* < 0.01), respectively). Further, cancer aggressiveness is clinically assessed morphologically by pathological grade system. High G2M score pancreatic cancer demonstrated a significantly high proportion of higher grade in both TCGA and GSE62452 cohorts (Figure 2B; *p* = 0.047 and *p* = 0.005, respectively). The high G2M score group was significantly associated with high proliferation score in the TCGA cohort (Figure 2C, *p* < 0.001), which was calculated using data from Thorsson et al. [18].

### 2.3. Clinical Characteristics of the High G2M Pathway Score Pancreatic Cancer Patients

Given that G2M scores reflect pancreatic cancer cell proliferation, we expected that high G2M score tumors would be associated with aggressive clinical characteristics. Table 1 shows the clinical characteristics (age at diagnosis, gender, race, primary tumor site, histological diagnosis, American Joint Committee on Cancer (AJCC) T-, N-, and M-category, and Stage) among low and high G2M score pancreatic cancer. Surprisingly, none of clinical features of aggressive cancer, such as larger primary tumor, lymph node metastasis, nor distant metastasis, demonstrated statistically significant distribution by the G2M score.

### 2.4. High G2M Pathway Score Tumor is Associated Not Only with KRAS and TP53 Gene Alteration but also with Their Signaling in Pancreatic Cancer

It is well known that *KRAS, TP53, Smad4* and *CDKN2A* genes are frequently altered and associated with poor prognosis in pancreatic cancer [19,20,21]. In the TCGA pancreatic cancer cohort, gene alteration of *TP53, KRAS, Smad4* and *CDKN2A* were found to be 41%, 23%, 21% and 13%, respectively (Figure 3A). Since alteration of these genes are known to contribute to poor prognosis, we hypothesized that G2M pathway score is associated with alteration of these genes and with their signaling. To test the hypothesis, we compared the percentage of patients with alteration of each genes between low or high G2M score groups using the TCGA cohort. Indeed, high G2M score tumor was significantly associated with a higher percentage of alteration in *KRAS* and *TP53* (Figure 3B, *p* = 0.003 and *p* = 0.026, respectively), but not with *Smad4* or *CDKN2A* (Figure 3B, *p* = 0.703 and *p* = 0.217, respectively). High G2M score tumor was also significantly associated with high gene expression of *KRAS* and *TP53*, which are reported to relate with each gene mutation (Figure 3C, *p* < 0.001 and *p* = 0.003, respectively), whereas there was no association with *SMAD4* and *CDKN2A* gene expressions. Lastly, we investigated the relationship between G2M score and the signaling of the above-mentioned gene alterations measured by each Hallmark set of GSVA. We found that high G2M score tumors significantly associated with less *KRAS* signaling down, elevated *P53* pathway, tumor growth factor (*TGF*)-β signaling, and *E2F* target score (Figure 3D; all *p* < 0.001).

### 2.5. The High G2M Pathway Score is Significantly Associated with High Fraction of T Helper Type 2 (Th2) Immune Cells

Previously, we reported that the high G2M score for breast cancer is associated with a characteristic tumor-immune microenvironment [17]. Since cancer cell proliferation and aggressiveness can be attenuated by immune cell infiltration in the tumor microenvironment [22], we expected that high G2M score tumors may have a characteristic immune cell infiltration profile. Utilizing the xCell algorithm, we found that pancreatic tumors with high G2M score were significantly associated with lower fraction of CD8 T cell, and M2 macrophage, and higher fraction of T helper type 1 (Th1) cells and T helper type 2 (Th2) cells in the TCGA cohort (Figure 4A). On the other hand, tumors with high G2M score were significantly associated with higher CD4 memory T cell and Th2 in the GSE62452 cohort (Figure 4A). Combined, high infiltration of Th2 cells alone consistently associated with G2M score high tumor in both cohorts. In order to further elucidate the relationship between the G2M score and cancer immunity, we further analyzed the association of G2M pathway score with several other scores: interferon (*IFN*)-γ and tumor infiltrating lymphocyte (TIL) regional fraction, lymphocyte infiltration score, and leukocyte fraction score, which were previously calculated by Thorsson et al. [18]. Lymphocyte infiltration signature score was significantly lower in the high-score G2M pathway tumor (Figure 4B, *p* = 0.008); however, the high G2M score tumor did not associate with any trend in any of the other scores.

### 2.6. High-Score G2M Pathway Pancreatic Cancer was Significantly Associated with Worse Survival Consistently in Multiple Cohorts

Given our results, we investigated the impact of G2M pathway score on pancreatic cancer patient prognosis. We found that high-score G2M pathway tumor was significantly associated with worse clinical outcome in disease-free survival (DFS), disease-specific survival (DSS), and progression-free survival (PFS), as well as in overall survival (OS) in the TCGA cohort (Figure 5; *p* = 0.018, *p* = 0.010, *p* = 0.036, and *p* = 0.006, respectively). This result was completely mirrored by the other pancreatic cancer cohorts, GSE62452 (*n* = 69, *p* = 0.001), and GSE57495 (*n* = 63, *p* = 0.038), which have gene expression data with OS (Figure 5).

### 2.7. High G2M Pathway Score Pancreatic Tumors were Significantly Associated with Worse Survival in Pathologically Margin-Positive Resection (R1 or R2), but not in Margin-Negative Resection (R0)

Our group and others have reported that margin-positive (R1 or R2 (R1/2)) resection of pancreatic tumors result in worse survival compared with margin-negative (R0) resection, and margin positivity is speculated to be due to aggressive cancer biology rather than surgical technique [4,5,6]. To this end, we hypothesized that high G2M tumors are more likely to have positive margin (R1/2) and worse survival. To test this hypothesis, the percentage of patients who achieved R0 resection was compared between low and high G2M score tumors; however, no statistically significant difference was observed in the TCGA cohort (Figure 6A, *p* = 0.102).

Next, we examined the patient survival (DSS and OS) separately in R0 and R1/2 groups. Interestingly, high G2M score tumors were not associated with worse survival in neither DSS or OS in patients who achieved R0 resection in the TCGA cohort (*p* = 0.236 and 0.118, respectively), whereas they were significantly associated with worse survival in both DSS and OS in patients who had R1/2 resection (Figure 6B, *p* = 0.008 and 0.006, respectively). This result led us to speculate that high G2M score tumors in the R0 resection group are biologically different from those in the R1/2 resection group. To this end, we performed the pathway analysis of high G2M score tumors in R0 and R1/2 groups and analyzed the percentage of each category of gene sets in the Hallmark collection with normalized enrichment score (NES) > 1.5 (Figure 6C). We found that pancreatic tumors with high G2M score in the R1/2 group were highly enriched with metabolism and cellular component gene sets, whereas tumors with high G2M score in the R0 group were enriched more immune-related (3-fold higher) and DNA repair-related (2-fold higher) gene sets (Figure 6C). This biological difference was even more clear when high G2M tumor-enriched gene sets were compared between R0 and R1/2 (Appendix A). All the cell proliferation-related gene sets (*P53* pathway, *MYC* targets v1 and v2, *E2F* targets, and Mitotic spindle) as well as cell proliferation-related metabolic (oxidative phosphorylation and glycolysis) and DNA repair (UV response up and DNA repair) gene sets were strongly enriched (NES > 1.55) to high G2M tumors in R1/2, but not in R0. On the other hand, immune response-related (*IFN-*γ response, inflammatory response, complement) were the most strongly enriched gene sets to high G2M tumors in the R0 group. These results implicate that G2M score is not a biomarker for R0 resection, however, it detects aggressiveness associated with R1/2 margin-positive pancreatic cancer.

Given these results, we sought whether the prognostic value of G2M pathway score is independent of other clinical and pathological features. This was accomplished using univariate and multivariate Cox regression analyses. We found that G2M score had a significant hazard ratio (HR) with grade (G3/G4 vs. G1/G2), AJCC T and N category, and resection status (R1/2 vs. R0) in univariate analysis of disease-specific survival (DSS) in the TCGA cohort (Table 2). Using multivariate analyses, we found that G2M score was prognostic, independent of other clinical factors in DSS (hazard ratio [HR] = 2.08, 95% confidence interval [CI] = 1.02–4.24, *p* = 0.045).

## 3. Discussion

In this study, a total of 390 pancreatic cancer patients were examined to investigate the relationship between G2M checkpoint activity with clinical features such as cancer aggressiveness, immune cell infiltration, and patient survival in pancreatic cancer. Previously reported G2M pathway score was used where high and low score groups were divided by median value. Our findings suggest that G2M score was elevated in cell proliferation gene sets such as *E2F* targets, MYC targets v1 and v2, and mitotic spindle formation in two large pancreatic cancer cohorts. This result was mirrored in clinical parameters: *MKI67* expression, pathological grade, and proliferation score. Independent of other factors, G2M score was predictive of prognosis such as pancreatic cancer survival, albeit not with immunity. Additionally, a high G2M score was also associated with increased mutation rates in *KRAS* and *TP53*, with mRNA expression being elevated for both genes. Other genes which showed increased expression included p53, *TGF*-β, and *E2F* pathways and a low score in the *KRAS* down pathway. Although G2M score did not correlate with the pathologically complete R0 resections, high G2M score tumors were significantly associated with worse DSS and OS after margin-positive R1 or R2 resections, but not after R0 resections. We further found that high G2M score tumors after R1 or R2 resection enriched more metabolic and cellular component-related gene sets compared to after R0 resection. To our knowledge, this is the first and currently the only study to use gene set expression data to examine the relevancy of G2M pathway activity in pancreatic cancer.

The impact of this study is that G2M score may have a clinical utility as a prognostic biomarker in pancreatic cancer patients after margin-positive resection.

In this study, none of clinical features of aggressive cancer, such as large primary tumor or lymph node metastasis of distant metastasis, demonstrated statistically significant association with G2M score, however, G2M score was associated with pathological grade. This indicates that G2M score may be more suitable for grasping the tumor aggressiveness compared to classic clinical features such as tumor size or lymph node metastasis that depend upon morphological analyses. Indeed, we found that R1/2 group high G2M tumors enriched cell proliferation-related gene sets stronger than the R0 group, whereas the R0 group high G2M tumors enriched immune response-related gene sets. The limited number of stage IV samples prohibited us to evaluate the association of G2M score with metastasis, but the GSEA results showed that the metastasis-related gene sets were not enriched in neither high nor low G2M score groups.

Gene alteration in the oncogene *KRAS* as well as tumor suppressor genes *TP53/p53*, *SMAD4/CPC4,* and *p16/CDKN21* are the most common and major driver genes in pancreatic ductal adenocarcinoma (PDAC) [23,24,25], which contribute to poor prognosis [19,20,21]. Specifically, it was found that patients with an un-altered form of *KRAS* or *CDKN21/p16* lived significantly longer than those harboring the mutations [26]. P16 controls the G1 to S cell cycle checkpoint by acting as an inhibitor of cyclin-dependent kinases and thus suppressing cell growth, and when the function of this important protein is suppressed by mutation, homozygous deletion, or hypermethylation, can cause aberrant cell growth [26,27]. *SMAD4*, is a key signaling protein in the *TGF*-β pathway, and plays a controversial role in the development of cancer [28,29,30]; however, it is known that the number of *SMAD4* mutations correlates with patient prognosis [26,31]. Expression of the tumor suppressor gene *TP53* was found to be significantly lower in PDAC tissues as compared to normal and benign tissues and expression levels significantly correlated with clinical outcomes, with lower *TP53* associated with increased risk of cancer development and worse survival.

Tumor immune microenvironment has been repeatedly demonstrated to be deeply involved in cancer progression by our group and others [32,33,34,35,36,37,38,39,40]. Recently, antitumor immunity was reported to correlate with increased cell cycle activity in pancreatic cancer [41,42]. We demonstrated that immune cell infiltration is higher with high G2M pathway score in breast cancer [17]. To this end, we expected high G2M score pancreatic cancer to demonstrate high immune cell infiltration; however, this turned out to not be the case given inconsistent immune cell composition in two cohorts. Our results implicate that each cancer type has unique biology and tumor immune microenvironment that cannot be easily generalized.

Currently, multimodality imaging is used to determine whether pancreatic cancer is indicated for surgical resection [43]. Pathological margin-negative R0 resection has better survival compared with margin-positive R1 or R2 resection [4,44,45]. In majority of the cases, it is not only surgical technique, but also the aggressive biology of pancreatic cancer result in positive margins (R1 or R2) after pancreatectomy [4]. R0 resection is often not achieved because pancreatic cancer is highly invasive. There is increasing evidence that cancer biology is the determining factor in achieving complete curative resection [43]. Interestingly, we found that high G2M score pancreatic cancer in R0 is biologically different compared from that in R1/2 in the current study. Our results align with the notion that there is a biological difference between highly proliferative pancreatic tumor in R0 and in R1/2 resection. Further investigation is warranted to determine whether this difference may translate into change in management of these tumors.

Targeted therapy against the cell cycle pathway or related genes is expected to improve survival in pancreatic patients [46,47]. *CDK1* regulation of this checkpoint is a key tumorigenic event, and it has been shown that the G2M pathway-related genes are a potential therapeutic target in pancreatic cancer [48,49,50]. *CDK4* and *CDK6* are two additional regulatory proteins which control the activity of cyclin D1, the activities of which mediate *Rb* and *E2F* and control the transition from G1 to S in the cell cycle. *CDK4* targeted therapy can inhibit the G1S progression and has been shown to control cancer development [51]. Furthermore, combinational inhibition of both *CDK4* and *CDK6* is expected to have an even greater benefit to the clinical outcomes of PDAC patients [52]. It has been shown that an approach which uses *CDK4/6* inhibitors after taxanes more effectively slows proliferation of PDAC cells [53]. Since G2M pathway score can accurately assess the cell cycle activity, we cannot help but speculate that the score may be useful as a biomarker for patient selection for *CDK4/6* inhibition for pancreatic cancer. We believe our results warrant further study to analyze the correlation between the score and the treatment effect in future.

This study still has some limitations. Although we have utilized multiple cohorts to validate these novel findings, it remains a retrospective study. A prospective study will be required in order to establish the G2M pathway score as a predictive biomarker. Another limitation is that none of the cohorts we analyzed have detailed information on systemic treatment and it is assumed that all the patients underwent “standard of care”. Thus, the changes and roles of G2M pathway score in tumor under chemotherapy remains unknown. The lack of treatment information is a major limitation in the study on pancreatic cancer, where systemic treatment has a critical impact on the outcome. Another limitation of the TCGA cohort is that they include a very low number of stage IV patients compared with the general public. We conducted our analyses including stage IV to maximize the power of the analyses. The current study analyzed surgically resected primary pancreatic cancer, which is less than 20% of all pancreatic cancer. Thus, our findings in unresectable and metastatic cancer may or may not be the same. Finally, it is known that a half of patients recur even after R0 resection, which implicates that G2M score does not reflect the behavior of micrometastasis given that there was no survival difference by score after R0 resection. Finally, experimental studies are needed to investigate the mechanism and applicability of our findings in the future.

## 4. Materials and Methods

### 4.1. Pancreatic Cancer Cohorts and Their Data

Transcriptomic profiling and clinical information of the TCGA pancreatic cancer (TCGA_PAAD) was downloaded from the Genomic Data Commons Data Portal (GDC). To date, we have published a number of reports that elucidated the clinical relevance of microRNA expressions [54,55,56] and gene expressions [37,38,39,57,58], and the underlying biology of pathological findings [59,60,61,62,63,64]. To maximize the number of pancreatic cancer samples to have the strongest power for statistical analyses, data of 176 patients who had pathological diagnosis of pancreatic cancer were included in the study. 82.4% of the samples were identified as pancreatic ductal adenocarcinoma. For the clinical staging system, we used the prior editions to the 7th edition of the American Joint Committee on Cancer (AJCC). Mutation data was downloaded from cBioportal as we described previously [65]. Gene Expression Omnibus (GEO) repository of the US National Institutes of Health provides normalized genomic and clinical data. The average value was used for genes with multiple probes. Gene expression data, which were transformed for log_2_, were used in all analyses as we have previously reported [6,17,36,58,66]. We used published data of Hussain et al. (GSE62452; tumor sample; *n* = 69, adjusted pancreases sample; *n* = 61) [67], Chen et al. (GSE57495; *n* = 63) [68], and Bocklandt et al. (GSE28746; tumor sample; *n* = 45, adjusted pancreases sample; *n* = 45) [69] to investigate the association of the G2M pathway scores with patient survival.

### 4.2. G2M Scoring Method

The gene set variation analysis (GSVA) algorithm [70] was utilized to measure the G2M score as the GSVA score for the “HALLMARK_G2M target” gene set of the MSigDb Hallmark collection [71] using the GSVA Bioconductor package (version 3.10), as we previously reported [17]. To divide low and high pathway score groups, the median value of each score within cohorts was used.

### 4.3. Gene Set Expression Analysis

Gene set enrichment analysis (GSEA) was a publicly available software (GSEA version 4.0) provided by the Broad Institute (Cambridge, MA, USA) [72]. For GSEA, we utilized Hallmark gene sets used for this study. We used the adjusted *p*-values (false discovery rate (FDR)) less than 0.25 for statistical significance, which was reported by GSEA software (Lava version 4.0).

### 4.4. Cell Composition Fraction Estimation

To compare the fraction of immune cells between groups, xCell score was calculated through transcriptomic data based on the xCell algorithm [73]. The xCell algorithm was used to estimate tumor-infiltrating immune cells in bulk tumors. Data was downloaded through the xCell website (https://xcell.ucsf.edu/), as we previously reported [6,17,36,66].

### 4.5. Statistical Analysis

R software (version 4.0.1, R Project for Statistical Computing) and Microsoft Excel (version 16 for Windows) were used in all data analysis and data plotting. One-way analysis of variance (ANOVA) and two-tail Fisher’s exact tests were used for analyses of comparisons between groups, as we described in each figure legends. For uni- and multi-variate analysis, Cox proportional hazard analyses were used to estimate HR, 95% CI, and *p*-value. *p*-value less than 0.05 was taken as statistically significant. Tukey-type boxplots show median and interquartile level values.

## 5. Conclusions

We found that the G2M pathway score identified poor survival in pancreatic cancer, particularly after margin-positive resection.

## Figures and Tables

**Figure 1 cancers-12-02871-f001:**
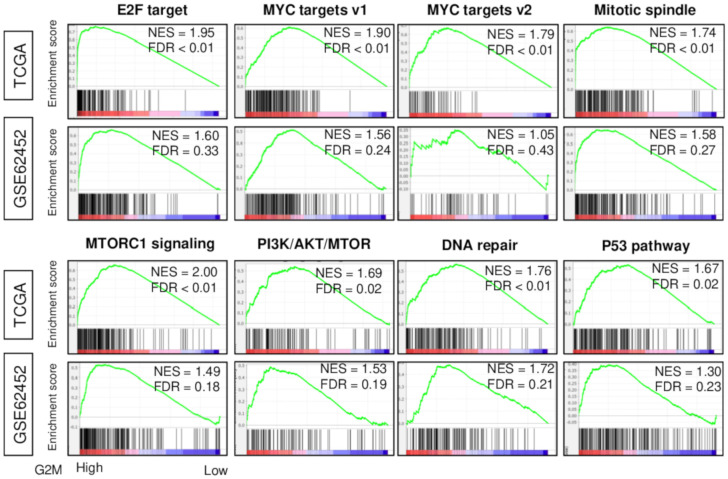
Hallmark gene sets that significantly enriched in high G2M pathway score pancreatic cancer. Gene set enrichment plots along with normalized enrichment score (NES) and false discovery rate (FDR) are shown for the eight (*E2F* target, *MYC* targets v1, *MYC* targets v2, Mitotic Spindle, *MTORC1* signaling, *PI3K/AKT/mTOR* signaling, DNA repair, *P53* pathway) gene sets that significantly enriched in tumors with high compared to low G2M pathway score in both TCGA and GSE62452 cohorts. The G2M score was calculated from tumor gene expression as the single-sample gene set variation analysis score for the Hallmark G2M gene set, and within-cohort median value was used to identify tumors with high and low scores. NES and FDR were determined with the classical GSEA method, where FDR < 0.25 is considered significant.

**Figure 2 cancers-12-02871-f002:**
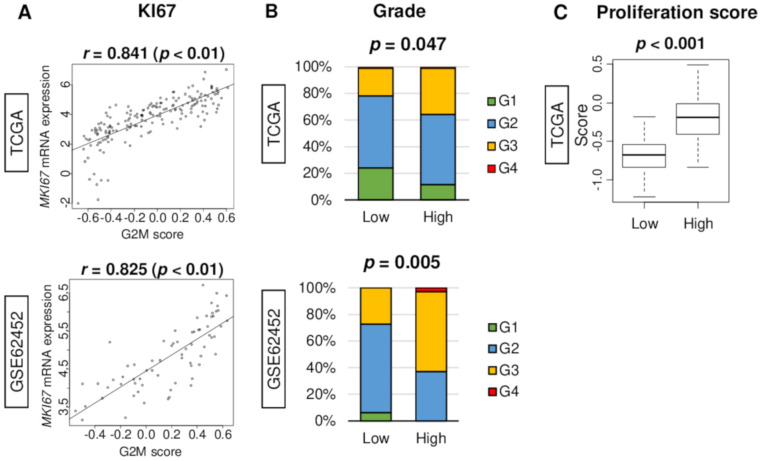
Association of G2M pathway score with clinical parameters of cancer cell proliferation in TCGA and GSE62452 cohorts. (**A**) Correlation between the G2M score and *MKI67* expression in both cohorts. *p-*value was analyzed with spearman *r* correlation. (**B**) Bar plots of the G2M score high and low groups demonstrate distribution of pathological grade: Grade 1 (G1, green), Grade 2 (G2, blue), Grade 3 (G3, yellow), and Grade 4 (G4, red) in the TCGA and GSE62452 cohorts. (**C**) Boxplot of proliferation score by low and high G2M score group in the TCGA cohort. High and low score groups were divided by the median value. Tukey-type boxplots show median and inter-quartile level values, and the analysis of variance (ANOVA) test is used to calculate *p-*values.

**Figure 3 cancers-12-02871-f003:**
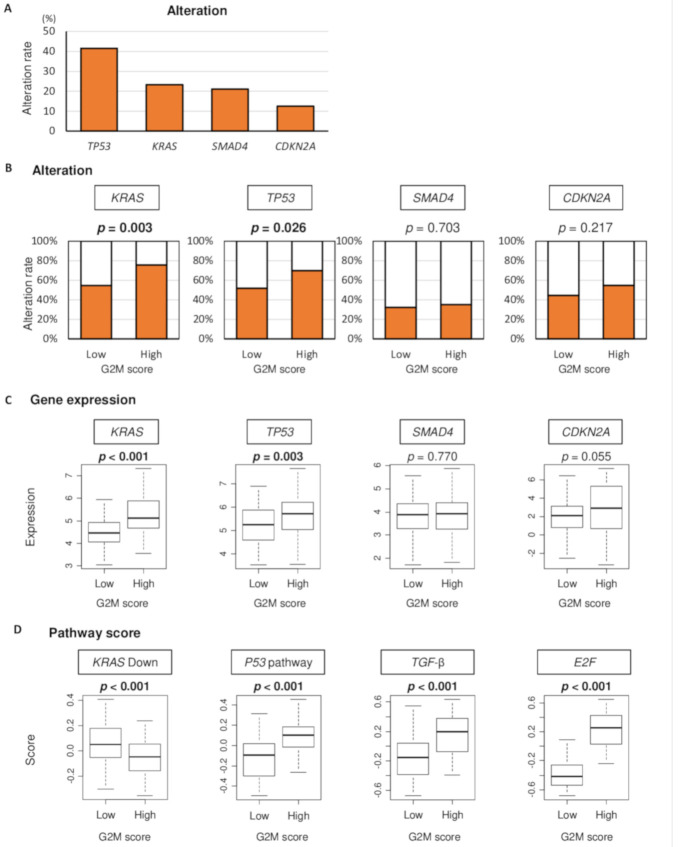
The association of G2M pathway score with alteration, expression, and signaling of *KRAS, TP53, SMAD4,* and *CDKN2A* genes in the TCGA cohort. (**A**) The percentage of patients with alteration of each gene in the TCGA pancreatic cancer cohort. (**B**) Percentage of patients with alteration of each genes in low or high G2M score groups. (**C**) Comparison of levels of *KRAS, TP53, SMAD4,* and *CDKN2A* gene expression by low and high G2M score groups. (**D**) Comparison of KRAS signaling up, *P53* pathway, *TGF*-β signaling, and *E2F* target score by low and high G2M pathway score. High and low score groups were divided by the median value. Tukey-type boxplots show median and inter-quartile level values, and the ANOVA test is used to calculate *p*-values.

**Figure 4 cancers-12-02871-f004:**
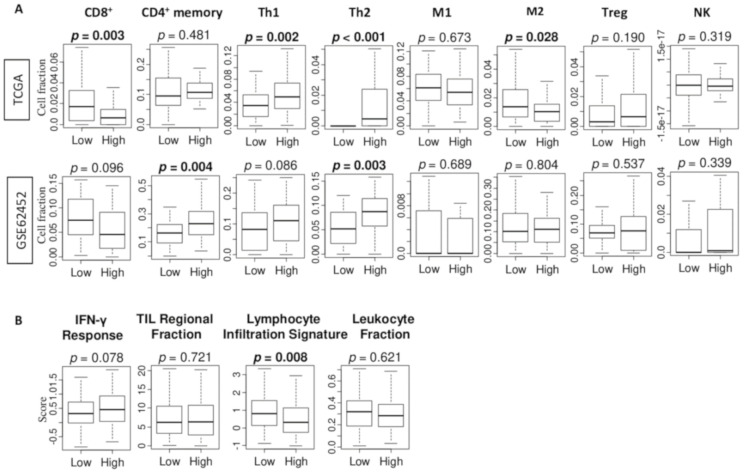
The association of G2M pathway score with tumor infiltrating immune cells estimated by xCell algorithm and immune function scores in the TCGA and GSE62452 cohorts. (**A**) Boxplots of the estimated amount of CD8 T cell, CD4 memory T cell, type 1 helper T cell (Th1) and type 2 helper T cell (Th2), M1 and M2 macrophage, regulatory T cell, and natural killer T cell by low and high G2M pathway scores. (**B**) Boxplots of scores: interferon (*IFN*)-γ response, tumor infiltrating lymphocyte (*TIL*) regional fraction, lymphocyte infiltration, and leukocyte fraction by low and high G2M pathway score. High and low score groups were divided by the median value. Tukey-type boxplots show median and inter-quartile level values, and the ANOVA test is used to calculate *p*-values.

**Figure 5 cancers-12-02871-f005:**
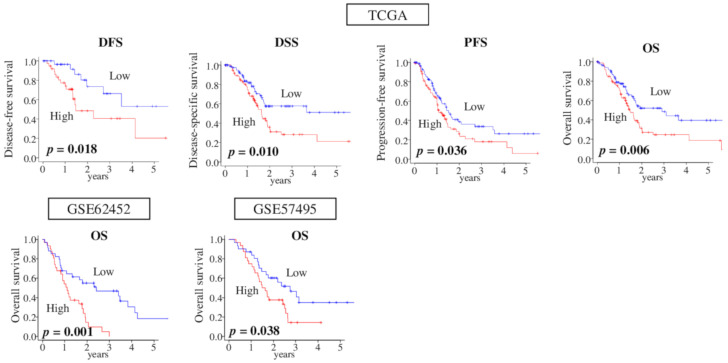
Association of G2M pathway score and pancreatic cancer patient survival in TCGA, GSE62452, and GSE57495 cohorts. Kaplan–Meier survival plots comparing low and high G2M score tumors for disease-free survival (*DFS*), disease-specific survival (*DSS*), progression-free survival (*PFS*), and overall survival (*OS*) in the TCGA (*n* = 176) cohort, and *OS* in GSE62452 (*n* = 69), and GSE57495 (*n* = 63) cohorts. High and low score groups were divided by the median value. Log rank test was used to compare between two groups with Kaplan–Meier survival curves and to calculate *p-*values.

**Figure 6 cancers-12-02871-f006:**
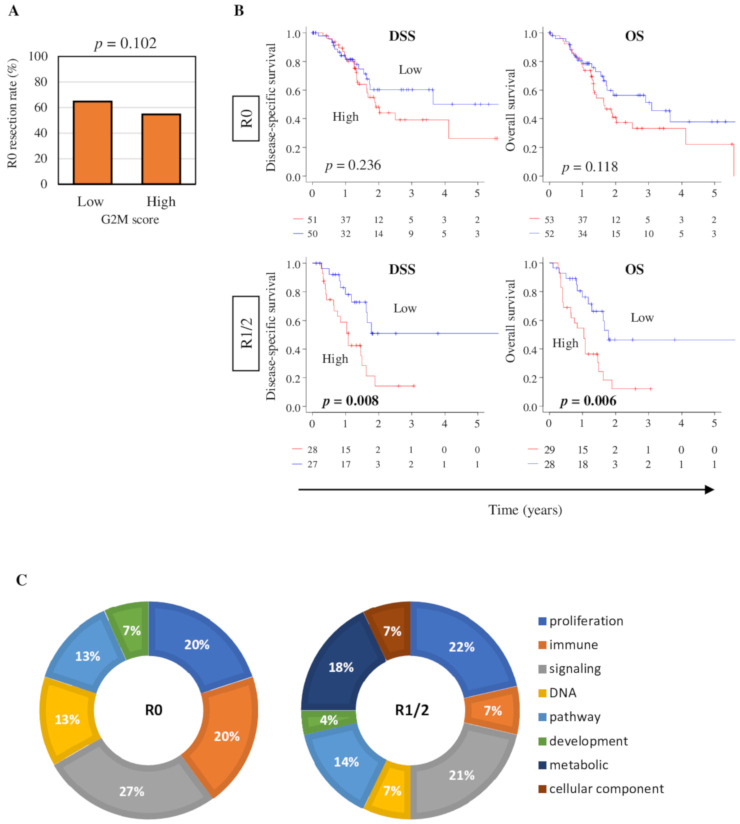
Association between G2M pathway score and resection status and respective survival in the TCGA cohort. (**A**) Achievement rate of R0 resection between low and high G2M pathway score in the TCGA cohort. *p*-value was determined by two-tailed fisher’s exact test. (**B**) Kaplan–Meier survival plots comparing low and high G2M score tumors in R0 or R1/2 resection groups, for OS and DSS in the TCGA cohort. High and low score groups were divided by the median value. Log rank test was used to compare between high and low score groups in Kaplan–Meier plots. (**C**) Pie charts of the categories of Hallmark gene sets for which significant enrichment (NES > 1.5 and FDR < 0.25) was observed in high G2M score compared to low G2M score tumors in R0 and R1/2 groups.

**Table 1 cancers-12-02871-t001:** Clinical characteristics of the low and high G2M score pancreatic cancer patients in The Cancer Genome Atlas (TCGA) cohort.

Clinical Variables	G2M-Low (*n* = 88)	G2M-High (*n* = 88)	*p*-Value
Age at diagnosis			0.950
Median	65.0	65.0	
IQR	57–74	58–71	
Gender			0.650
Male	50 (56.8%)	46 (52.3%)	
Female	38 (43.2%)	42 (47.7%)	
Race			1.00
White	78 (88.6%)	77 (87.5%)	
Other	10 (11.4%)	11 (12.5%)	
Primary site			0.680
Head	66 (75.0%)	71 (80.7%)	
Body/Tail	15 (17.0%)	13 (14.8%)	
Unknown	7 (8.0%)	4 (4.5%)	
Histology			0.429
PDAC	70	75	
Others	18	13	
Grade			0.047
G1	21 (23.9%)	10 (11.4%)	
G2	47 (53.4%)	46 (52.3%)	
G3	18 (20.5%)	30 (34.1%)	
G4	1 (1.1%)	1 (1.1%)	
Unknown	1 (1.1%)	1 (1.1%)	

AJCC T-category			
T1	4 (4.5%)	3 (3.4%)	0.524
T2	14 (15.9%)	9 (10.2%)	
T3	66 (75.0%)	75 (85.3%)	
T4	2 (2.3%)	1 (1.1%)	
Unknown	2 (2.3%)	0 (0.0%)	
AJCC N-category			
N−	26 (29.5%)	23 (26.1%)	0.612
N+	58 (65.9%)	64 (72.7%)	
Unknown	4 (4.6%)	1 (1.1%)	
AJCC M-category			
M−	37 (42.0%)	41 (46.6%)	1.00
M+	2 (2.3%)	2 (2.3%)	
Unknown	49 (55.7%)	45 (51.1%)	
AJCC Stage at diagnosis			0.540
I	13 (14.8%)	8 (9.1%)	
II	69 (78.4%)	76 (86.4%)	
III	2 (2.3%)	1 (1.1%)	
IV	2 (2.3%)	2 (2.3%)	
Unknown	2 (2.3%)	1 (1.1%)	

Resection status			0.102
R0	57 (64.8%)	48 (54.6%)	
R1/2	23 (26.1%)	34 (38.6%)	
Unknown	8 (9.1%)	6 (6.8%)	

AJCC, American joint Committee on Cancer; IQR, Interquartile range; PDAC, pancreatic ductal adenocarcinoma.

**Table 2 cancers-12-02871-t002:** Survival analyses of the G2M pathway score and other factors in the TCGA cohort.

TCGA (DSS)	Univariate		Multivariate	
	HR (95% CI)	*p-*Value	HR (95% CI)	*p-*Value
Age	1.02 (1.00–1.04)	0.109		
Gender (Male vs. Female)	0.77 (0.48–1.22)	0.267		
Race (Caucasian vs. other)	1.35 (0.45–2.83)	0.420		
Primary tumor site (Head vs. Body/Tail)	1.69 (0.86–3.31)	0.126		
Grade (G3/4 vs. G1/2)	3.28 (1.19–9.07)	0.022 *	0.71 (0.12–4.27)	0.706
AJCC-T category (pT3/4 vs. pT1/2)	3.08 (1.33–7.14)	0.009 *	2.46 (0.59–10.19)	0.215
AJCC-N category (N+ vs. N-)	2.58 (1.39–4.81)	0.003 *	1.93 (0.96–3.86)	0.064
AJCC-M category (M+ vs. M-)	1.21 (0.29–5.09)	0.799		
Resection (R1/2 vs. R0)	1.82 (1.11–2.98)	0.017 *	1.58 (0.96–2.61)	0.072
G2M score	2.65 (1.36–5.16)	0.004 *	2.08 (1.02–4.24)	0.045 *

AJCC, American joint Committee on Cancer; CI, confidence interval; DSS, disease-specific survival; HR, hazard rate. * *p* < 0.05.

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
