# Peer review of "High G2M Pathway Score Pancreatic Cancer is Associated with Worse Survival, Particularly after Margin-Positive (R1 or R2) Resection"

_cancers, 2020, doi:10.3390/cancers12102871_

Round 1
Reviewer 1 Report
The manuscript by Oshi et al. is well designed and of interest. I highly recommend its publcation. Minor revisions: please, revise the use of bold type and the alignment of the paragraphs.
Reviewer 2 Report
The manuscript is interesting because it analyzes the G2M pathway in pancreatic cancer and its relation with clinical characterictics and survival of patients. This is a retrospective study.
The introduction is poor. It is not adequately explained how the G2M score is obtained and how its cut-off point has been established, which is fundamental for understanding the rest of the manuscript.
It is referred that G2M pathway score may reflect tumor cell proliferation, immune cell infiltration and that it has been related to survival in breast cancer; available data in breast and other cancers should be detailed. Also, it is important to delve into the reason for the choice of the study of this score in pancreatic cancer.
The material and methods are insufficiently explained and incorrectly ordered, making it difficult to present the results.It should explain the differences between GSVA and GSEA, which algorithms are used as Cell algorithm, the score of the lymphocyte infiltration signature, etc, in addition, of which patient cohorts are analyzed (for example, in section 2.7 of the results, the survival in the GSE 78229 cohort is analyzed; n=50, to which no reference had been made previously).
Table 1, which shows the characteristics of the patients (age at diagnosis, sex, race, location of primary tumor, histological diagnosis, AJCC TNM stage) indicates that in each group according to the high or low G2M pathway score there are 88 patients, respectively, however, the sum of patients according to the characteristics does not match. It is not indicated if there are patients whose clinical characteristics are unknown or what happens. Patients with tumors other than pancreatic adenocarcinoma are included and this should also be pointed out why it is done and what significance it has.
Table 1 should include the histological grade and resection status, which are then analyzed in relation to the score.
The number of patients with metastatic tumors is very low (4.6% of the total), so the conclusions of the studies carried out on this sample will be less certain. It should assess whether it should have selected patients more homogeneous in their characteristics or choose subgroups of patients to study the G2M pathway score.
The results are interesting, but their relevance is scarcely discussed. For example, regarding clinical features, none of clinical features of aggresive cancer, such as large primary tumor, lymph node metastasis or distant metastasis demonstrated statistically significant distribution by G2M score, however, G2M pathway score was associated to aggressiveness parameters as histological grade. These results should be analyzed and explained.
The G2M score is correlated with measures of patient survival, although it is unknown what treatments they received (chemotherapy, radiotherapy, etc.).
The section of the results 2.5, which refers to the association of G2M pathway score high with KRAS and TP53 gene alteration and with their signaling in pancreatic cancer should be explained in more detail.
The results of section 2.8 should be related to those obtained in section 2.4 and explained together.
In the discussion the results should be further evaluated. The conclusions mainly refer to the group of resected patients and it is speculated whether G2M score high tumors in R0 resection group are biologically different from that in R1/2 resection group, but it should be analyzed whether this is really demonstrated in this study and how it could be demonstrated, in addition to its clinical significance. In addition, you must explain how important the rest of the results obtained are.
Reviewer 3 Report
Major concerns:
- The work as presented lacks of significant novelty and impact
- Most, if not all, the data are mere correlation between G2M signature and known drivers of pancreatic cancer
- The association between G2M signature to more proliferative/advanced cancers, with KRAS/p53 mutations is likely reflection of the tumor stage rather than predictive of anything
- As highlighted by the authors, the lack of drug treatment data and any association with G2M signature contributes to the lack of interest
- Some of the present data could be validated in vitro using several biochemical assays, and in vivo in genetically engineered mouse models of pancreatic cancer, PDXs and human samples
Minor concerns:
- Attention to the format of the gene names
- Revise the style and make it uniform
Round 2
Reviewer 1 Report
All my concerns have been adressed.
Reviewer 2 Report
The manuscript has been improved and the importance of its results is better reflected in the current form. Thank you for the corrections made.
Reviewer 3 Report
The author improved the quality of the manuscript.
Minor concern:
- Remove the bold font from line 65 to line 95 and verify the character size
- Insert a space at line 113 between “cancer” and “gene set”.
- Uniform the size of the figure legends
- Line 128 write Figure 2A instead of Figure. 2A
- Line 131 write Figure 2B instead of Figure. 2B
- Line 144 remove the double dots at the end of the sentence
- Line 204 remove the double dots at the end of the sentence
- Line 223 write Figure 4B instead of Figure 4
- Line 235 remove the double dots at the end of the sentence
- Line 486 write survival. Instead of survival;
